# Draft Genome Sequence of *Priestia* sp. Strain TSO9, a Plant Growth-Promoting Bacterium Associated with Wheat *(Triticum turgidum* subsp. *durum)* in the Yaqui Valley, Mexico

**DOI:** 10.3390/plants11172231

**Published:** 2022-08-28

**Authors:** Maria Edith Ortega-Urquieta, Valeria Valenzuela-Ruíz, Debasis Mitra, Sajjad Hyder, Nabil I. Elsheery, Pradeep Kumar Das Mohapatra, Fannie Isela Parra-Cota, Sergio de los Santos-Villalobos

**Affiliations:** 1Instituto Tecnológico de Sonora, 5 de Febrero 818 sur, Ciudad Obregon 85000, Sonora, Mexico; 2Department of Microbiology, Raiganj University, Raiganj 733 134, Uttar Dinajpur, West Bengal India, India; 3Department of Botany, Government College Women University Sialkot, Sialkot 51310, Pakistan; 4Agricultural Botany Department, Faculty of Agriculture, Tanta University, Tanta 31527, Egypt; 5Campo Experimental Norman E. Borlaug, Instituto Nacional de Investigaciones Forestales, Agrícolas y Pecuarias, Ciudad Obregon 85000, Sonora, Mexico

**Keywords:** whole-genome sequence, PGPB, genomic, microbial inoculant

## Abstract

Strain TSO9 was isolated from a commercial field of wheat (*Triticum turgidum* L. subsp. *durum*) located in the Yaqui, Valley, Mexico. Here, the genome of this strain was sequenced, obtaining a total of 5,248,515 bp; 38.0% G + C content; 1,186,514 bp N50; and 2 L50. Based on the 16S rRNA gene sequencing, strain TSO9 was affiliated with the genus *Priestia*. The genome annotation of *Priestia* sp. TSO9 contains a total of 147 RNAs, 128 tRNAs, 1 tmRNA, and 5512 coding DNA sequences (CDS) distributed into 332 subsystems, where CDS associated with agricultural purposes were identified, such as (i) virulence, disease, and defense (57 CDS) (i.e., resistance to antibiotics and toxic compounds (34 CDS), invasion and intracellular resistance (12 CDS), and bacteriocins and ribosomally synthesized antibacterial peptides (10 CDS)), (ii) iron acquisition and metabolism (36 CDS), and (iii) secondary metabolism (4 CDS), i.e., auxin biosynthesis. In addition, subsystems related to the viability of an active ingredient for agricultural bioproducts were identified, such as (i) stress response (65 CDS). These genomic traits are correlated with the metabolic background of this strain, and its positive effects on wheat growth regulation reported in this work. Thus, further investigations of *Priestia* sp. TSO9 are necessary to complement findings regarding its application in agroecosystems to increase wheat yield sustainably.

## 1. Introduction

As the world population is expected to increase to 9.8 billion by the year 2050 [1], the global community faces the current challenge of increasing food production to satisfy the required demand. Wheat is the third most cultivated cereal in the world, having annual production of approximately 734 million tons [2]. According to OECD-FAO, by the year 2027, its demand will grow to 833 million tons, which means that it is projected to rise by 13% above the baseline annual production [3]. Nevertheless, several abiotic and biotic factors negatively affect worldwide wheat production, i.e., extreme temperatures, soil salinity, ultra-violet radiation, phytopathogens, and the bioavailability of nutrients [4,5,6]. Thus, annual wheat production requires the application of high amounts of chemical fertilizers (∼400 kg N ha^−1^) [7], a situation that has intensified unsustainable agricultural practices, placing food security and the preservation of the environment at risk [8]. Although chemical fertilizers are created to improve the growth and productivity of crops, their chemical composition, and their excessive use, have repercussions not only in terms of decreasing food quality but also in accelerating environmental deterioration [9]. The use of agrochemicals strongly impacts air, water, and soil pollution—for example, (i) releasing greenhouse gases (GHG) (approximately one quarter of all anthropogenic GHG emissions) that significantly contribute to climate change [10], (ii) causing the eutrophication of water reservoirs through agricultural runoff [11,12,13], and (iii) the accumulation of heavy metals in agricultural soils [14]. Specifically, chemical fertilizers strongly alter the functions and properties of the soil, both physically, causing variations in its texture, bulk density, infiltration rate, hydraulic conductivity, and porosity, among others, and chemically, by deteriorating its structure, affecting its nutrient content and its cation exchange capacity, altering its pH, and causing biological damage in the microbial community, composed of bacteria, fungi, algae, archaea, protozoa, and nematodes [15,16]. This imbalance has led to soil degradation, salinization, loss of fertility, reductions in organic matter content, and the incidence of pests, which cause a reduction in wheat yield and high economic costs [8,17].

Thus, due to the negative impacts of the conventional, non-sustainable agricultural practices, it is of paramount importance to develop and implement innovative strategies to sustainably increase the productivity of agroecosystems through the use of beneficial microorganisms. A promising strategy is the identification and application of plant growth-promoting bacteria (PGPB), which are a group of beneficial bacteria that stimulate growth and nutrition, increasing the productivity and health of crops [8]. In this way, the bioprospection of these microorganisms is a suitable strategy for exploring the microbial ecology in current agroecosystems and determining their agro-biotechnological potential, since several studies have demonstrated that bacterial diversity is positively correlated with the macronutrient content and the soil fertility.

Through complex interactions, PGPB use various mechanisms that contribute to the solubilization of organic and inorganic phosphates and other nutrients, produce secondary metabolites such as siderophores, which act as iron chelators [18], and produce phytohormones such as auxins, gibberellins, and cytokines, among others. On the other hand, PGPB can indirectly promote plant growth through the stimulation of systematic resistance, competition for space and niche, and the production of antibiotics and lytic enzymes, among others, which together function as bio-controllers of phytopathogens. These mechanisms favor the use of PGPB as microbial inoculants (biofertilizers and/or biopesticides), as the association that involves soil–plant–bacteria is an excellent alternative to the partial or complete replacement of chemical fertilizers [19,20,21].

Therefore, this work is a robust genomic study of strain TSO9, which was isolated from a commercial field of wheat (*Triticum turgidum* L. subsp. *durum*) located in the Yaqui Valley, Mexico. Based on the raw data obtained from DNA sequencing and through the use of bioinformatics tools, the verification of contamination, assembly, alignment, and annotation of the bacterial genome of interest was carried out, contributing to the identification of genes involved in plant–bacteria interactions related to growth promotion, which was validated by several metabolic tests and interaction strain TSO9 in wheat plants.

## 2. Materials and Methods

### 2.1. Bacterium Culture Conditions

The bacterial strain TSO9 was isolated from the soil of a commercial wheat field located in the Yaqui, Valley, Mexico (27.3692°, 110.3886°). For this, a 10 g composited soil sample was homogenized with 90 mL sterile (121 °C and 15 psi for 15 min) distilled water, and the serial dilution (1:10) method was used up to 10^−6^. One mL of this was spread on a Petri dish containing nutrient agar (NA), in triplicate, and incubated for 2 days at 28 °C [22]. After incubation, strain TSO9 was characterized and purified based on its morphological traits, such as cell and colonial shape, color, elevation, and opacity, and then it was cryopreserved at −80 °C by using nutrient broth (NB) culture medium with glycerol (30%), in the Colección de Microorganismos Edáficos y Endófitos Nativos (COLMENA, www.itson.edu.mx/COLMENA, (accessed on 11 January 2022) [19,21].

### 2.2. Metabolic Characterization

Strain TSO9 was metabolically characterized regarding the most studied biochemical activities associated with plant growth promotion.

Production of indole acetic acid (IAA) was assessed as mentioned by de los Santos et al. [23]. First, 1 mL (1 × 10^6^ cells/mL) was inoculated in 10 mL of NB supplemented with 100 mg/L of tryptophan at 30 ± 2 °C for 5 days, in a rotary shaker at 120 rpm. After incubation, the production of IAA was determined by spectrophotometry assays according to Glickmann and Dessaux [24].Phosphate solubilization. Strain TSO9 was spot-inoculated, in triplicate, using 10 µL (1 × 10^6^ cells/mL) in Petri dishes containing Pikovskaya agar [25] and incubated for 7 days at 28 ± 2 °C [26]. The presence of a transparent halo around the inoculated colony was recorded as a positive result.Siderophore production. Chrome Azurosol S (CAS) agar was prepared from four solutions, which were sterilized separately before mixing. The culture medium was spot-inoculated in triplicate using 10 µL (1 × 10^6^ cells/mL) and incubated for 7 days at 28 ± 2°C [27]. The presence of a yellow–orange halo around the inoculated colony was recorded as a positive result.Abiotic stress. Here, 1 × 10^6^ colony-forming units (CFU) of strain TSO9 were spot-inoculated on Petri dishes containing NA as a culture medium and supplemented with (i) sodium chloride to determine saline stress and (ii) Polyethylene Glycol 6000 (10%, −0.84 mPa) to determine hydric stress; and (iii) the inoculum was incubated at a temperature of 43.5 °C for 3 days, to determine the thermal stress. The control treatment was conducted by spot-inoculating 1 × 10^5^ CFU of strain TSO9 containing only NA and incubating it at 28 °C [22]. The growth of strain TSO9 under these conditions was recorded as tolerance to abiotic stress.Biocontrol. A dual confrontation assay against the wheat phytopathogen *Bipolaris sorokiniana* TPQ3 was carried out. A volume of 10 µL of 1 × 10^5^ conidia/mL was spot-inoculated in the center of a Petri dish (8 cm in diameter) containing potato dextrose agar, and then 10 µL of 1 × 10^6^ CFU of strain TSO9 was spot-inoculated at four equidistant points (2 cm of distance), in triplicate, around *B. sorokiniana* TPQ3, and the sample was incubated for 5 days at 28 °C [6]. The growth inhibition of the phytopathogen indicated biocontrol by the strain TSO9.Bacteria—Wheat plants interactions were assessed under a greenhouse assay. The growth promotion traits of strain TSO9 were analyzed through a greenhouse assay where fifteen wheat seeds (var. CIRNO C2008) per treatment were germinated on Petri dishes containing agar–agar (8 g L^−1^). Then, the wheat seedlings (7 days post-germination) were transplanted in pots containing 1.5 kg of non-sterilized soil and were inoculated; for this, the bacterial strain TSO9 was grown in 30 mL of sterile nutrient broth contained in a Falcon tube (50 mL) and incubated for 2 days at 28 °C and 180 rpm. Then, the bacterial culture was centrifuged at 3600 g for 10 min and the obtained pellet was washed twice and re-suspended in sterile distilled water. The optical density (630 nm) of strain TSO9 was adjusted to 0.5 (1 × 10^8^ CFU mL^−1^). Thus, 5 mL (5 × 10^8^ CFU) of this strain was inoculated on the wheat’s rhizosphere, and in the negative control, the cell suspension was replaced with 5 mL sterile distilled water. The greenhouse assay was carried out for 3 months, under the climatic conditions of the Yaqui Valley (13 h of darkness at 14 °C, 2 h of light at 18 °C, 7 h of light at 25 °C, and 2 h of light at 18 °C). Plant biometric parameters such as leaf number, stem diameter, stem height, root length, and plant dry weight were evaluated as described by Valenzuela-Aragon et al. [22]. Results were reported as mean values and percentages, calculated using the following formula: [(Treatment value − Control value)/Control value] × 100.

All data were expressed as the means of studied replicates. Significant differences were analyzed by the one-way analysis of variance (ANOVA) test and Tukey–Kramer test (*p* < 0.05), using Statgraphics Centurion XVI.II.

### 2.3. Genomic Analysis

High-quality genomic DNA was extracted from a fresh culture of strain TSO9, which was grown in NB (24 h at 32 °C, using an orbital shaker at 121 rpm, obtaining 1 × 10^6^ CFU/mL), and following the protocol described by Valenzuela-Aragon et al. [22]. DNA sequencing was performed by using the Illumina MiSeq platform (2 × 300 bp) (Illumina, San Diego, CA, USA). Next-generation sequencing (NGS) library preparation was carried out by using the TruSeq DNA Nano Kit for Illumina^®^ Platforms, according to the manufacturer’s instructions.

Furthermore, the quality of the obtained reads was analyzed by FastQC version 0.11.5 [28]. Meanwhile, Trimmomatic version 0.32 [29] was used to remove adapter sequences and low-quality bases. Subsequently, a de novo assembly was generated by SPAdes version 3.14.1 [30], using the “--careful” parameter for error correction in reads. The assembled contigs were ordered by Mauve Contig Mover version 2.4.0 [31], using the reference genome of *Priestia megaterium* ATCC 14581^T^ (GenBank accession number GCA_000832985.1). In addition, plasmid detection was carried out by PlasmidFinder 2.0 [32]. Finally, a 16S rRNA-based-phylogenetic tree was constructed by CLC Sequence Viewer version 8.0 (CLC bio A/S, Qiagen, Aarhus, Denmark), using the neighbor-joining construction model, and *Bacillus vallismortis* DV1F-3T (genebank accession number JH600273) was used as an outgroup.

### 2.4. Genome Annotation

The genome annotation of *Priestia* sp. TSO9 was developed by the Rapid Annotation Using Subsystem Technology (RAST) server version 2.0 (http://rast.nmpdr.org) (accessed on 11 January 2022) [33], using the RASTtk pipeline based on the PathoSystems Resource Integration Center (PATRIC) [34]. In addition, a second annotation platform named CGView Server beta was also used, which has recently been renamed Proksee (https://proksee.ca/) (accessed on 11 January 2022) [35], providing the Rapid Prokaryotic Genome Annotation (Prokka) [36], which generated a circular chromosome map of *Priestia* sp. TSO9, including the CDS, tRNAs, rRNAs, and guanine–cytosine (GC) skew content.

### 2.5. Genome Mining

To identify the biosynthetic potential of strain TSO9, its genome was submitted to the web server Antibiotics & Secondary Metabolite Analysis Shell (AntiSMASH) 6.0 (https://antismash.secondarymetabolites.org/) (accessed on 11 January 2022), under the “relaxed” parameter, which allows the rapid identification, annotation, and analysis of gene clusters related to the biosynthesis of secondary metabolites, non-ribosomal peptide synthetases, polyketide synthases, type I and II polyketide synthases, lasso peptides, and antibiotic oligosaccharides, among others [37].

## 3. Results

### 3.1. Morphological and Metabolic Characterization

Strain TSO9 presented a morphological characterization of Gram-positive rod-shaped cells, and a white, circular, flat, and opaque colony. In addition, this strain presented functional traits associated with plant growth promotion, such as phosphorus solubilization (54 ± 1.0%) and tolerance to thermal (118 ± 3.1%), saline (72 ± 1.3%), and hydric (113.6 ± 1.9%) stress. On the other hand, the biometric parameters of inoculated wheat plants were measured, showing significant (*p* < 0.05) increments vs. the non-inoculated treatment in the leaf number (68.3%) and the stem diameter (87.9%); however, stem height (12.1), root length (13.5%), and plant dry weight (7.9%) did not show significant differences (Table 1). Finally, the capacity of strain TSO9 for siderophore and IAA production, as well as biocontrol against *B. sorokiniana* TPQ3, were not observed.

### 3.2. Genomic Analysis

The bacterial DNA was sequenced, obtaining a total of 3,807,277 total paired-end reads (2 × 300 bp). After assembly, the draft genome of *Priestia* sp. TSO9 presented 60 contigs in 16 scaffolds (minimum of 1455 bp and maximum of 4,984,774 bp), resulting in 5,248,415 bp; 38.0% G + C content; 1,186,514 bp N50; and 2 L50. In addition, plasmids were not detected in this genome. Then, the 16S rRNA gene analysis showed 100% similarity to *Priestia megaterium* NBRC 15308^T^, 99.86% to *P. aryabhattai* B8W22^T^, and 98.95% to *P. flexa* NBRC 15715^T^ (Table 2), which was confirmed by the 16S rRNA-based-phylogenetic tree (Figure 1). However, due to the phylogenomic similarity to closely related *Priestia* species, strain TSO9 was taxonomically affiliated to the genus *Priestia.*

### 3.3. Genome Annotation

The genome annotation developed by RAST predicted a total of 147 RNAs and 5623 CDS, distributed into 332 subsystems (Figure 2). The subsystems (Appendix A) with the most significant presence of coding DNA sequences (CDS) were (i) amino acids and their derivatives (388 CDS); (ii) carbohydrates (318 CDS); (iii) protein metabolism (224 CDS); iv) cofactors, vitamins, and prosthetic groups (163 CDS); and v) nucleosides and nucleotides (103 CDS). Moreover, the genome of this strain presented CDS related to plant growth promotion, such as (i) virulence, disease, and defense (57 CDS), i.e., resistance to antibiotics and toxic compounds (34 CDS), invasion and intracellular resistance (12 CDS), and bacteriocins and ribosomally synthesized antibacterial peptides (10 CDS), which exert their antibacterial effects and inhibit the growth of closely or non-closely related bacterial strains; (ii) iron acquisition and metabolism (36 CDS), i.e., siderophores (17 CDS), which act as iron chelators and reduce the availability of phytopathogenic microorganisms that depend on this element; and (iii) secondary metabolism (4 CDS), i.e., auxin biosynthesis, which plays an important role in shaping plant organogenesis, tropic responses, and plant morphogenesis in general. Furthermore, subsystems related to bacterial resilience for designing promising agricultural bioproducts were identified, such as the stress response (65 CDS), i.e., osmotic stress (14 CDS) and oxidative stress (20 CDS). In addition, complementing the results, the circular chromosome map based on Prokka predicted a total of 5395 CDS, 130 tRNAs, and 1 tmRNA (Figure 3).

### 3.4. Genome Mining

The genome mining carried out by the web server AntiSMASH 6.0 resulted in the identification of eight regions, where only regions 1, 3, and 8 presented clusters of biosynthetic genes such as surfactin (13%), carotenoid (50%), and the antibiotic microccocin (8%); these results indicate that, due to the lower similarity percentage (<70%) of biosynthetic gene clusters, *Priestia* sp. TSO9 does not possess clusters associated with a biocontrol capacity, which is in line with the lack of this ability in the confrontation assays, mentioned above.

## 4. Discussion

Now more than ever, we are seeing how indispensable the proper bioprospection of beneficial bacteria is for agricultural bioproducts, mitigating the negative human health and environmental effects in agrosystems caused by conventional agricultural practices. In the era of fast and accessible sequencing, there has been an increase in microbial taxon identification and reclassification, as is the case for the genus *Priestia*. This genus of Gram-positive, mostly rod-shaped bacteria in the Bacillaceae family was recently reclassified by Gupta et al. [38]. Although plasmids are common in the genus *Priestia* [39], 25% of the reported *Priestia* strains do not have plasmids, as found in *Priestia* sp. TSO9.

The genus *Priestia* has been previously reported as a plant growth-promoting bacterium in tomatoes [40], increasing its photosynthetic rate and fruit weight per plant, as well as increasing the lycopene content and total carotenoids. It has also been reported to promote wheat growth [41], being able to improve germination. Moreover, the signaling pathways of auxin and ethylene have been attributed to the plant growth promotion traits, relating these pathways to root development [42]. In this manner, the genome of strain TSO9 presents genes (19) related to plant growth promotion through auxin biosynthesis. Auxins are an important group of hormones for plant growth and development, which play an important role in shaping plant organogenesis, tropic responses, and plant morphogenesis in general [43]. IAA is the most commonly found and physiologically active phytohormone in plants, where the shoot apical meristems of plants produce IAA in the form of diffusible auxins and can be found in almost all plant tissues [44]. It has been reported that more than 80% of rhizospheric bacteria can synthesize and release auxins [45]. However, this strain does not produce indoles, which may be due to the lack of necessary conditions or the absence of genes from the full metabolic pathway for indole production. Thus, more studies are needed to determine *Priestia* sp. TSO9′s ability to produce IAA.

On the other hand, *Priestia* sp. TSO9 showed the ability to solubilize phosphorus 54 ± 1.0%, which could potentially be associated with the following genes involved in phosphorous metabolism, which were found in its genome: IPP, phoP, EPP, ptsA, Oprp, and ET1, where 25 CDS from the strain TSO9 genome were identified (Appendix A). Phosphorus is an important nutrient for plant growth and development, playing a vital role in metabolic processes, energy transfer and storage, plant photosynthesis, respiration, the formation of the cell membrane, glycolysis, and enzyme activities [46]. Moreover, phosphorus is the major yield-limiting plant nutrient in arid and semi-arid soils [47]. Despite its limited (<1%) solubilization in soil and, thus, plant absorption [48], plant growth is dependent on phosphorus assimilation. In plants, it increases root and stem development, improves seed formation, and increases crop maturity and nitrogen fixation. In this way, the inoculation and presence of phosphorus-solubilizing bacteria in soils, such as *Priestia* sp. TSO9, is crucial for the bioavailability of phosphorus through the release of organic acids and enzymatic activity [49].

*Priestia* sp. TSO9 also presented 61 genes related to siderophore production (Appendix A). Siderophores are low-molecular-weight chelating agents (200–2000 Da) [50]. Iron is one of the most essential elements required for the development and normal functioning of plants and microorganisms; although iron is abundantly available in the soil, it is present in complex insoluble forms, where microorganisms play a major role by making it soluble and chelating it from available complex organic or inorganic iron, and, therefore, providing the required iron for plants, thus resulting in plant growth promotion [18,51]. However, although *Priestia* sp. TSO9 contains genes related to siderophore production, identified through the RASTtk platform, it does not produce them in CAS medium [22]. Moreover, genes related to siderophore production were not detected by the antiSMASH server. Thus, this leads us to infer that there may be an absence of important genes from the full metabolic pathway for its production, where in-depth specific analyses are needed to identify the determinant gene sets for *Priestia* sp. TSO9’s siderophore production.

Other traits of interest in PGPB are stress resistance abilities, so that it may increase its competence level in the applied habitat. *Priestia* sp. TSO9 showed the ability to tolerate thermal (118 ± 3.1%), saline (72 ± 1.3%), and hydric stress (113.6 ± 1.9%) stress. In this manner, 14 CDS were identified to be related to osmotic stress and thus thermal, saline, and hydric stress, such as OsmY, OmpA, Prop(OH)2, glycerol, and aqua genes (Appendix A).

Although *Priestia* sp. TSO9 does not show biological control activity against *Bipolaris sorokiniana*, *Priestia* species are commonly identified as biocontrol agents, and their potential as such may not be disregarded for other phytopathogens. Thus, the multiple anti-pathogenic mechanisms of *Priestia* species include (i) the production of iron-chelating siderophores as identified in the inhibition of brown root rot against *Fomes lamaoensis*; (ii) the synthesis of cell wall-degrading enzymes, peroxidase, phenylalanine ammonia-lyase, chitinase, and β-1,3-glucanase [52], as observed against *Rhizoctonia solani*, the causative agent of damping off [53]; (iii) the production of volatile compounds, as described against the aflatoxin-producing *Aspergillus flavus* on rice grains [54]; and (iv) lastly, through the inactivation of acyl-homoserine lactones associated with plant-protective, quorum-quenching activity to the quorum sensing of plant pathogenic bacteria [39].

In addition to the previously mentioned PGPB traits that *Priestia* sp. TSO9 possesses, plant growth promotion effects are significantly (*p* < 0.05) positive in wheat, being able to increase the leaf number by 68.3% and the stem diameter by 87.9%, in comparison to non-inoculated wheat plants. Moreover, *Priestia* species are widely reported as PGPB; for example, *Priestia megaterium* has been reported as PGPB in tomato [55], but has also been reported to promote other plants’ growth, such as wheat [41] and kale [56]. Thus, *Priestia* sp. TSO9 contains other genes of agricultural interest, such as bacteriocins and ribosomally synthesized antibacterial peptides, with 14 genes related to this function (10 CDS) (Appendix A). The capacity to produce antimicrobial peptides is widespread among Gram-positive bacteria [57]. These substances are directed against competitive microorganisms, and thereby generate a selective advantage for their producers. Bacteriocins comprise a heterogeneous family of small, ribosomally synthesized, proteinaceous molecules with strong antimicrobial activity [58]. In addition, these antimicrobial peptides have a bacteriostatic or bactericidal spectrum of activity that is mainly directed against bacteria closely related to the producing strain [59] but may also act against other non-related bacteria [60,61].

## 5. Conclusions

The need for an increase in food production due to current and future demand and vulnerable food security has led producers to the excessive use of chemical fertilizers, which not only poses a risk to food security but also affects the quality of air, water, and soil, negatively impacting the environment and the economy of farmers. Thus, it is essential to carry out genomic studies aimed at bioprospecting promising bacteria that can be applied to the soils, and, through their variety of reported and unknown mechanisms, promote plant growth and contribute to the development of cutting-edge tools that have a positive impact on agriculture, offering an alternative to the use of chemical fertilizers and then contributing to the reduction of the adverse effects of non-sustainable agricultural practices. In this context, *Priestia* sp. TSO9 contains a great number and diversity of genes that support its metabolic background and positive effects as a plant growth-promoting bacterium. Nevertheless, further research is necessary to explore the diverse functional activities and genes of strain TSO9 to design a bacterial inoculant for wheat production sustainably.

## Figures and Tables

**Figure 1 plants-11-02231-f001:**
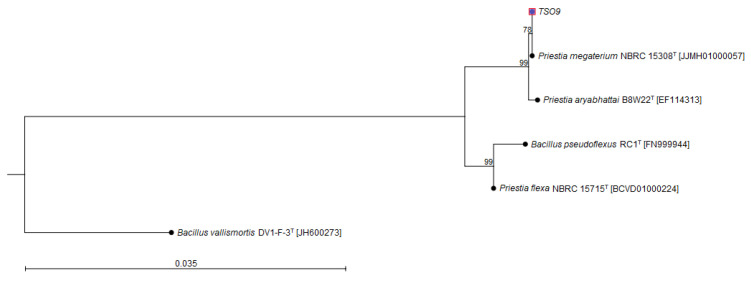
Phylogenetic relation between strain TSO9 and closely related species (based on the 16S rRNA gene): *Priestia megaterium* NBRC 15308T (JJMH01000057); *P. aryabhattai* B8W22T (EF114313); *P.* flexa NBRC 15715T (BCVD01000224), and *Bacillus pseudoflexus* RC1T (FN999944). *B. vallismortis* DV1-F-3T (JH600273) was used as an outgroup. This relation was constructed by CLC Sequence Viewer v 8.0.0 with the nucleotide distance measure Jukes–Cantor, and the neighbor-joining construction model (based on 1000 bootstrap replications). Scale bar (0.035) represents the number of nucleotide substitutions per site.

**Figure 2 plants-11-02231-f002:**
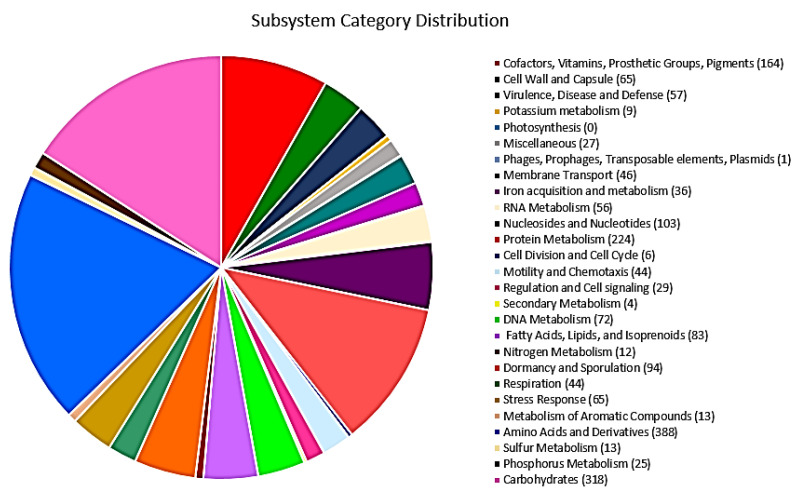
Pie chart of the subsystem category distribution of CDS from *Priestia* sp. TSO9 was constructed by RAST server version 2.0. CDS: 5623; CDS in subsystems: 1981; and subsystems: 332.

**Figure 3 plants-11-02231-f003:**
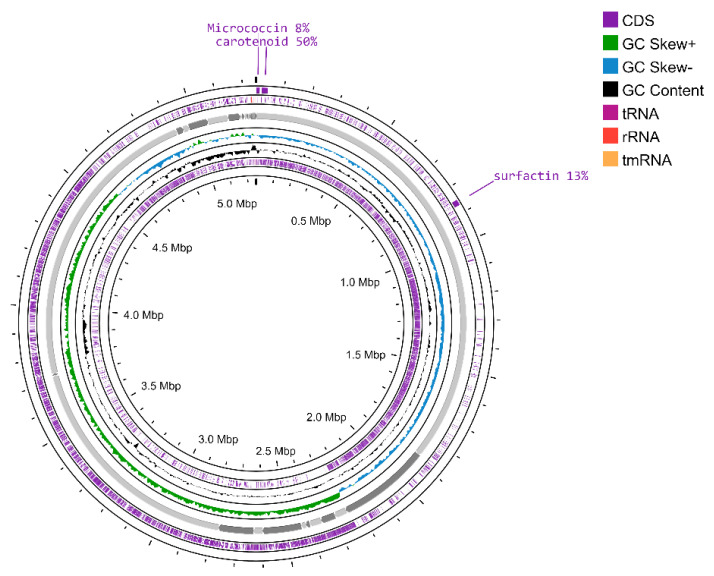
Circular chromosome map of *Priestia* sp. TSO9, which includes the distribution of CDS, tRNAs, rRNAs, and GC content skew, was created by CGView Server beta.

**Table 1 plants-11-02231-t001:** Wheat plants’ growth promotion by the inoculation of the strain TSO9.

Parameter/Treatment	Leaf Number	Increment (%) vs. Control	Stem Diameter, cm	Increment (%) vs. Control	Stem Height, cm	Increment (%) vs. Control	Root Length, cm	Increment (%) vs. Control	Plant Dry Weight, g	Increment (%) vs. Control
Non-inoculated (control)	9.9 ± 2.9	-	6.1 ± 1.9	-	42.2 ± 5.9	-	31.9 ± 2.8	-	2.1 ± 0.7	-
Inoculated (strain TSO9)	16.8 ± 3.2 *	68.3	11.5 ± 2.5 *	87.9	47.3 ± 4.6	12.1	36.2 ± 3.4	13.5	2.3 ± 0.5	7.9

Asterisks (*) indicate statistically significant differences between inoculated and non-inoculated treatments, according to Tukey–Kramer test (*p* = 0.05). Means (*n* = 15).

**Table 2 plants-11-02231-t002:** 16S rRNA-based similarity of strain TSO9.

Taxon Name	Strain	GenBank Accession Number	Similarity (%)
*Priestia megaterium*	NBRC 15308^T^	JJMH01000057	100
*Priestia aryabhattai*	B8W22^T^	EF114313	99.86
*Priestia flexa*	NBRC 15715^T^	BCVD01000224	98.95
*Bacillus pseudoflexus*	RC1^T^	FN999944	98.72

## Data Availability

The draft genome sequence has been deposited in DDBJ/ENA/ GenBank under accession number JAJEKD000000000. The version described in this paper is the first version, JAJEKD000000000, under BioProject number PRJNA772765 and BioSample number SAMN22416249. The raw data files may be consulted under the following accession number SRR16918074.

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
