# Peer review of "Draft Genome Sequence of Priestia sp. Strain TSO9, a Plant Growth-Promoting Bacterium Associated with Wheat (Triticum turgidum subsp. durum) in the Yaqui Valley, Mexico"

_plants, 2022, doi:10.3390/plants11172231_

Round 1

Reviewer 1 Report

The manuscript introduce us to a new, possibly, high performant plant growth promoting bacteria.

The introduction is clear and easy to read. However, a better highlight of the need to discover new plant growth promoting bacteria for agriculture should be included.

Lines 69, 70: ”bacteria” instead of ”bacterias”

In Materials and Methods section:

Line 112: ”Strain TSO9 was inoculated” would be better if said ” ...spot inoculated”

The same comment for Lines 118 and 132.

Lines 136-144: 

1. The soil was autoclaved?

2. When the biometric measurements were done, at the end of the 3 month period? 

3. How the bacteria was inoculated?

4. When the bacteria was inoculated?

5. How many plants were used?

For more clarity these information should be added in the method. 

Results section:

Line 186: How the percentage were calculated? According to Materials and Methods section the determination were qualitatively done and not quantitatively. For more clarity, please add the information in Materials and Methods section.

The Discussion section is clear and easy to follow.

Conclusions

The conclusions should be improved by highlighting more the possible impact of the bacteria on agriculture.

Author Response

Thank you for revising our manuscript. All your comments were included in the manuscript, which improved its quality.

All corrections were highlighted in blue in the revised version of our manuscript.

The introduction is clear and easy to read. However, a better highlight of the need to discover new plant growth promoting bacteria for agriculture should be included.

R: Thank you for this suggestion, the requested information was added.

Lines 69, 70: ”bacteria” instead of ”bacterias”

R: We apologize for this mistake, this correction was made.

Line 112: ”Strain TSO9 was inoculated” would be better if said ” ...spot inoculated”

R: Thank you, the correction was made as suggested

The same comment for Lines 118 and 132.

R: The inoculation method was specified as suggested, thank you.

Lines 136-144: 

* The soil was autoclaved?

R: No, for this specific assay the soil was not sterilized to simulate the natural soil conditions in the field.

* When the biometric measurements were done, at the end of the 3 month period? 

R: Yes, the plant biometric traits were measured 3 months after plant inoculation.

* How the bacteria was inoculated?

R: Thank you for this improvement, this information was included in the revised version of our manuscript Strain TSO9 was inoculated on wheat’s rhizosphere.

* When the bacteria was inoculated?

R: Strain TSO9 was inoculated at the seedling stage of the wheat plant, 7 days post germination.

* How many plants were used?

R: Fifteen plants per pretreatment were used.

Line 186: How the percentage were calculated? According to the Materials and Methods section the determination was qualitatively done and not quantitatively. For more clarity, please add the information in the Materials and Methods section.

R: Thank you for your comment, the assay was done quantitively. The percentage values were calculated using the following formula: [(Treatment value – Control value)/ Control value] * 100. This information was included in the revised version of the manuscript.

The conclusions should be improved by highlighting more the possible impact of the bacteria on agriculture.

R: Thank you, the conclusion was improved as suggested.

Reviewer 2 Report

In this paper the Authors present the draft genome of Priestia sp.strain TSO9. The DNA sequencing, annotation and analysis are well described and interpretated.  I havea concern about the reported isolation of TSO9. The Authors describe the purification of strain TSO9 from a native soil (rhizosphere?) sample based on colony morphology which is an unlikely method to isolate a PGPB from the vast diversity of the soil microbiota.  On the other hand they cite a paper by their research groupnusing "plant-assisted selection" of wheat  PGPB in Reference 22.  Does not strain TSO9 derive from this study? Moreover, are not plant growth promotion data for TSO9 described in the latter publication?  The PGP data in the present manuscript are very scanty without primary data in graph or table format, and plant wet and dry weight not determined.  In the introduction the Authors amply discuss PGPB as alternatives to chemical nitrogen fertilizers but this  would be a trait of diazotrophs exclusively and is not addressed with regard to strain TSO9.  No measurements of biological nitrogen fixation are carried out nor the concentration of nitrogenous compounds in the seeds of inoculated versus uninoculated plants determined.  The latter is a general criticism of publications on PGPB or PGPR organisms. Is PGP measured on plant growth regulation related to wheat yield or seed N-content?  Are there nif/fix gene related ORF's in the draft DNA sequence?  

Strain TSO9 does not appear to synthesize auxins or siderophores, nor directs biocontrol activity, although ORF's related to these very important properties of PGPB are found in the draft sequence.  So are the PGP effects solely due to stress-related functions and phosphorus solubilization?  It seems that strain TSO9 is not a very good PGPB as compared to other PGPB or PGPR organisms described.

Strain TSO9 is shown to be of the Priestia genus, members of which have been studied for PGP characteristics and sequenced. Thus the analysis of the TSO9 sequence and metabolic properties is not necessarily very novel.

In summary, in spite of the fact that the determination of the draft sequence of Priestia sp. strain TS09 and its annotation are well described in this paper, and are in principle publishable per se, the biological (metabolic) studies are not very complete and not very encouraging for further in depth studies or possible utilization in the field 

Author Response

Thank you for revising our manuscript. All your comments improved its quality.

All corrections were highlighted in yellow in the revised version of our manuscript.

I have a concern about the reported isolation of TSO9. The Authors describe the purification of strain TSO9 from a native soil (rhizosphere?) sample based on colony morphology which is an unlikely method to isolate a PGPB from the vast diversity of the soil microbiota. 

R: Thank you for this comment and we apologize for the confusion. This study began with the conventional (the serial dilution method) isolation and purification of microorganisms from the soil in the Yaqui Valley, Mexico. Thus, as you mentioned, several bacterial colonies were observed in the Petri dishes; however, we decided to put attention and study one strain (TSO9) due to it was the most abundant colony on the dishes, and we hypothesized a potential functional role of this strain in the agroecosystem. Then, as we reported in this manuscript, the following steps were to characterize metabolically this strain, evaluate its effect on wheat morphology, and sequence its genome. All these approaches were carried out to explore its potential use as an active ingredient in the formulation of a bacterial inoculant for this geographical region and those having similar agricultural management, soil, and climate conditions. This development of this inoculant is still on the way.

On the other hand, they cite a paper by their research group using "plant-assisted selection" of wheat PGPB in Reference 22.  Does not strain TSO9 derive from this study? Moreover, are not plant growth promotion data for TSO9 described in the latter publication? 

R: We appreciate this comment, strain TSO9 was isolated for the current study, for this reason, all information about its isolation and characterization is reported here. Once this strain was isolated and some metabolic functions were determined for this study, this strain was used as a control in the manuscript published by Valenzuela-Aragón et al. [22], while the genome sequencing was carried out and more information was obtained from this strain for the current manuscript. The submission of the current work was delayed due to the sequencing and bioinformatics analysis being stopped by the COVID-19 pandemic. However, that time was used for learning about the previously published manuscript, and based on this information and the obtained genome sequence we were able to improve the protocols for detecting wheat growth promotion by strain TSO9, i.e. wheat plant - strain TSO9 interaction assay was modified increasing the evaluation period, as well as using native soil without sterilization, among other. Thus, the current manuscript complements and improves the findings reported by Valenzuela-Aragón et al. [22].

The PGP data in the present manuscript are very scanty without primary data in graph or table format, and plant wet and dry weight not determined. 

R: Thank you very much for this important suggestion to improve the quality of our manuscript. The data were detailed and included in Table 1 for the following parameters: Leaf number, Stem diameter, Stem height, Root length, and Plant dry weight. Unfortunately, we did not measure the Plant wet weight, for this reason, this value was not included in the table.          

In the introduction, the Authors amply discuss PGPB as alternatives to chemical nitrogen fertilizers but this would be a trait of diazotrophs exclusively and is not addressed with regard to strain TSO9.  No measurements of biological nitrogen fixation are carried out nor the concentration of nitrogenous compounds in the seeds of inoculated versus uninoculated plants determined. The latter is a general criticism of publications on PGPB or PGPR organisms. Is PGP measured on plant growth regulation related to wheat yield or seed N-content?  Are there nif/fix gene related ORF's in the draft DNA sequence?  

R: We agree with your suggestion, thank you. The introduction was improved according to your comments; thus, the wide description of nitrogen fertilizer was eliminated and included information about the benefits of PGPB. On the other hand, as you mentioned, this strain does not have the presence of nif/fix genes. Thus, as we indicated before, the introduction was modified.

Strain TSO9 does not appear to synthesize auxins or siderophores, nor directs biocontrol activity, although ORF's related to these very important properties of PGPB are found in the draft sequence.  So are the PGP effects solely due to stress-related functions and phosphorus solubilization?. It seems that strain TSO9 is not a very good PGPB as compared to other PGPB or PGPR organisms described.

R: Thank you for this comment. We are very excited about the biological traits of this strain. As we reported in the manuscript, the inoculation of this strain to wheat plants under greenhouse conditions showed promising beneficial effects. On the other hand, in the genome of strain TSO9, we can find genes involved in important pathways of plant growth promotion, but these metabolic traits were not detected. This could be due to in vitro we were not able to simulate the optimal conditions for inducing the biosynthesis of these metabolites and/or this strain is not able to synthesize these compounds due to a mutation in the annotated and putative genes, and/or the observed wheat growth promotion is due to others unexplored mechanisms. Now, we are starting some studies for deciphering the main mechanisms of bioactivity of strain TSO9, we hope to get these results in the coming year.

Strain TSO9 is shown to be of the Priestia genus, members of which have been studied for PGP characteristics and sequenced. Thus the analysis of the TSO9 sequence and metabolic properties is not necessarily very novel. In summary, in spite of the fact that the determination of the draft sequence of Priestia sp. strain TS09 and its annotation are well described in this paper, and are in principle publishable per se, the biological (metabolic) studies are not very complete and not very encouraging for further in depth studies or possible utilization in the field 

R: We appreciate your comment. We agree with the fact of other strains of the Priestia genus have been characterized and sequenced; however, TSO9 is the first strain of this genus isolated, characterized, and sequenced in the Yaqui Valley, Mexico, the Birthplace of the Green Revolution. In this region, the agricultural practices are very intensive, and soil (organic matter = 0.5%, electric conductivity ~ 2.0 ds m-1, pH ~ 8.5, etc) and climatic (Tmin ~ 4oC, Tmax ~ 45oC, rainfall of 300 mm a year, etc.), so PGPB used for moving forward to sustainable agriculture needs to be highly adapted to the agricultural management, soil and climate conditions. For this reason, we have reported this manuscript to increase the knowledge about a native and well-adapted promising PGPB for the Yaqui Valley, and other regions worldwide having similar actual conditions or those expected due to climate change.

In the revised version of this manuscript, we have included the requested additional information to improve its quality and enhance the promising correlation between metabolic and functional traits and the genome annotation. We continue working on this strain at the metabolic, ecological, and genomic levels to identify the main action mechanisms involve in wheat growth promotion. Thus, the publication of this manuscript will help us to establish a strong background for the coming findings.

Round 2

Reviewer 2 Report

The revised version of the paper now includes more data on the plant tests and in its present form is acceptable for publication.